# The (Processed) Vegetal Body and Blood of the Markan Messiah

**Dong Hyeon Jeong** 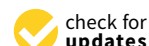

Department of Religion, Rutgers University, New Brunswick, NJ 08901, USA; dj353@religion.rutgers.edu

**Abstract:** Although the Eucharist is attested four times (Matthew 26:26–29; Mark 14:22–25; Luke 22:15–20; 1 Corinthians 11:23–26) in the New Testament, only two (Luke and 1 Corinthians) out of the four instances bespeak of commemorating this event ("Do this in remembrance of me"). Limiting the discussion to Mark's iteration of this event, Mark's version does not command to remember; rather he focuses on the ontological ("This is my body/blood"). This paper follows Stephen D. Moore's vegetal reading of the Johannine Jesus (*Gospel Jesuses and Other Nonhumans*) that invites and acknowledges the animacy of the vegetal in affectively re-engaging the identity of the messiah. That is, (processed) plants/food are not there just to be symbolically equated with the body and blood of the messiah. They re-animate and re-define the nature of messiahship. This paper utilizes Jane Bennett's vital materialism, Bruno Latour's actants, and Michael Marder's vegetality in arguing that Mark vegetally reconfigures the ontology of the messiah in the Eucharist/Last Supper scene (14:22–25). Deleuze and Guattari's concept of assemblage will assist in elaborating how the vegetal could dismantle anthropocentric understanding of ontology. By doing so, this paper opens up the possibility to reimagine a messiah who finds his identity with the vegetal or those that are considered dispensable.

**Keywords:** Eucharist; Gospel of Mark; vegetality; vital materialism; Colonialism; race and ethnicity; assemblage; actants

---

## 1. Tracing Liberatory Manifestations of the Eucharist

The Eucharist (also known as Communion), which means "thanksgiving," is a sacrament[1] performed by Christians as a remembrance of the Last/Lord's Supper (1 Corinthians 11:23–25; Matthew 26:26–29; Mark 14:22–25; and Luke 22:14–20). Jesus and his twelve disciples partook in this Supper right before Jesus was crucified. In this meal, Jesus broke the bread and gave the wine to his disciples as manifestations of his upcoming sacrifice for the other. Aside from Matthew and Mark, 1 Corinthians and Luke have Jesus commanding his disciples to commemorate/repeat this event as a way to remember him. Such command solidifies the role of the Eucharist as a sacramental, integral part of Christianity. Moreover, the Eucharist acts not just as an identity marker for Christianity, but it also represents the encounter and nearness of God, awaiting eagerly the unveiling of God's grace in the world.

One of the most crucial moments in this Supper is Jesus's signification of his body with the bread, and his blood with the wine. The interpretations of this signification vary because it hinges upon the understanding of the relationality between the divine and the matter: Are Jesus' body and blood literally the bread and the wine, or are they just symbolic? Among many possibilities, this article

---

[1] Sacrament is a term for Christian ceremonies that reflect an outward sign of inward grace of God.

focuses on the interpretation of the Eucharist from the intersections of ecological and (human) labor justice. Ecologically conscientious Christian communities argue that the extraordinary symbolisms imputed on the Eucharist should be (re-)rooted on the ordinary/material: Plants, land, water, and human labor. This imputation is grounded on the ordinary because it finds the presence of the divine in and with all of creation. The Eucharistic entanglement of the divine with all creation comes from the proposition that, as Margaret Scott elaborates, "the earth is a sacred reality" (Scott 2009, p. 59). This entanglement does/should not only reside in the symbolic and/or liturgical spectacle, as Monika K. Hellwig cautions: "The Eucharist has been a spectacle, something that people came to view, to view with reverence, with hope, gratitude and contrition, but nevertheless to view, standing outside the action" (Hellwig 1976, p. 3). This entanglement participates in reconfiguring the Eucharist from its alienating performativity to a more liberatory communal engagement. This liberatory call to action is a response to the intersectional mandate to care for the earth and the liberatory demands of the poor and the vulnerable. Scott echoes this rootedness with her incisive critique and challenge in understanding the Eucharist: "How do we make our eucharistic celebrations more alive and relevant, more about real people and not reduced to a private spiritual experience, lost in rubrics?" (Scott 2009, p. vi).

Andrea Bieler and Luise Schottroff also echo this liberatory understanding of the Eucharist by defining the Eucharist as a resurrection meal. Bieler and Schottroff find in the Eucharist redemption of those who are victims of systemic terror, malnutrition, and hunger (Bieler and Schottroff 2007, p. 3). Using the hermeneutics of sacramental permeability and eschatological imagination, Bieler and Schottroff express their view of the early church's understanding of the Eucharist as a "foretaste of resurrection while social conflicts related to poverty, slavery, and privilege—experiences of the powers of death—were not suppressed" (Bieler and Schottroff 2007, p. 6). Responding to anti-black sentiments and white supremacy still lingering in churches, Katie Walker Grimes also argues that the Eucharist cannot be an innocuous event performed in churches detached from the socio-ethical issues plaguing the church. Although Grimes did not work with new (or vital) materialism, Grimes admonishes that "we become what we eat, but food also becomes what it is eaten by" (Walker Grimes 2017, p. 201). If we partake the bread and the wine with hatred for the other, then the bread and the wine of the Eucharist becomes as such. In other words, the very consecrated body and blood of Jesus become affected by the socio-ethical issues of our time.

In this regard, this article follows Scott, Bieler and Schottroff, and Grimes' liberatory approaches by interpreting the Eucharist with vital materialism. That is, I argue that this entanglement between the divine and the material should include the reconfiguration of the ontology of the (Markan) Jesus. Most ecojustice understanding of the Eucharist finds the divine in all creation, thereby hyphenating the ontology of the creation with the divine. Yet, this hyphenation seems to avoid the reconfiguration of the messiah's ontology, keeping him essentially pure and untouched by the *ontos* of the created. Limiting my discussion with the Markan messiah, a vital materialist reading of Mark 14:22–25 opens up the possibilities for the Markan messiah to be ontologically reconfigured by the other. As Mark 14:22–25 has the Markan messiah signify ("is") his own body with the bread and the wine, this article argues that this signification is not just symbolic. Rather, it provokes an ontological reconfiguration in which the "human" and the "divine" are porously blurred by the nonhuman (bread and wine). In other words, the Markan Jesus is an assemblage of the human and the (processed) vegetal.

Reflecting further on the liberatory approach of this article, making the Eucharist alive and relevant traces its roots to Paul's teaching that acknowledges the entanglements of the consecration of the Eucharist with caring for the other. In 1 Corinthians 11:23–34, Paul indicted the church of Corinth because they celebrate the Eucharist perfunctorily or without fully understanding the socio-ethical demands of this ritual. They consecrate the bread and the wine but neglect to eat together, an act that reverts to oppressive social hierarchies. Interpreting Paul's indictment as liturgy-in-praxis is also reflected in *Didascalia Apostolorum*. The bishops are to care for the "the orphans and widows . . . in the likeness of the altar" (Connolly 1969, p. 88). Although Paul did not explicitly mention the need to care

for the earth in 1 Corinthians 11:23–24, the implications of his socio-ethical approach to the Eucharist creates spaces for further reflections on what it means to care for the other.

The cleric Bartolomé de las Casas echoes Paul's understanding of the Eucharist. Before he championed the plight of the poor, de las Casas accompanied European Christian colonizers in the conquest of Cuba. He spent at least twelve years participating in the violence against the poor of the Caribbean, having his own slaves to work for him. A propitious event in April 1514, however, changed de las Casas from the oppressor to the prophet of the oppressed. Diego Velásquez asked de las Casas to preach due to expediency and shortage of priests. Coincidentally, the passage he preached on that day was Ecclesiasticus 34:18–22. As he prepared his sermon, v.21 changed his heart: "The bread is life to the destitute, and it is murder to deprive them of it." The passage opened his eyes to the oppressive structures he himself is an active participant and beneficiary. He found himself unable to celebrate the Eucharist faithfully until the oppressive structures of his community could be eradicated. So, de las Casas freed his slaves first, then continued his prophetic work in Cuba, Santo Domingo, Spain, and even in West Indies.[2] Enrique Dussel finds de las Casas' change of heart as a reflection on the importance of intersecting the Eucharist with justice:

> Bread is the product-food for the alienated-poor who are forced to work, but do not consume, their life is objectivized in the product but it does not come back to them in the form of life-consumption. When bread is not life to the poor, the poor die. (Dussel 2003, p. 46)

Dussel implies that consecration is liberation. For the peasant Galilean Jesus to commemorate the bread as his body is to ground the Eucharist with the bodies of the oppressed. Reflecting on de las Casas' change of heart and the materiality of Eucharist, Mayra Rivera echoes Dussel's statement: "An elemental materiality connects the bodies of workers with shared bread, with consecrated bread. These are not arbitrary metaphors—bread is produced by the labor of human hands and the fecundity of the earth" (Rivera 2015, p. 23). The bread is not just there to be broken and eaten perfunctorily. They are the "real" bread, product of someone's hand. My addition to this discussion intersects the gift of life given by the plants, and the divine's ontological offering to become fluidly reconfigured with the struggles of the people and the earth.

From a Roman Catholic perspective, Izunna C. Okonkwo argues that understanding the Eucharist from the intersections of eco-justice and caring for the oppressed has been one of the central concerns of the Roman Catholic church since (or even before) the Second Vatican Council. Church documents such as "The Constitution on the Sacred Liturgy" (*Sacrosanctum Concilium*), "Pastoral Constitution on the Church" (*Gaudium et Spes*), and "Dogmatic Constitution on the Church" (*Lumen Gentium*) provide evidence to this claim (Okonkwo 2013, p. 527). Okonkwo argues that the divine's decision to be transubstantially present in what the humans have made from the earth (bread and wine) is a matter of great significance not just because the Eucharist transforms/invites the recipients to become positive changemakers of this world, but it also highlights the divine's decision to "remain and be remembered" (Okonkwo 2013, p. 536) by the creation. Although he does not engage the Eucharist with vital materialism, Okonkwo's exposition on the Eucharist invites a philosophical engagement in reconfiguring the ontology of the messiah as someone who deliberately transgresses his identity by assembling with the nonhumans.

From the Roman Catholic Church to Protestant perspectives, Catherine Keller voices her discontent with many Protestant denominations' decision to perform the Eucharist by abandoning the material and ecological significations substantiation provide.[3] As Keller states, "We might have cut off some access to the materialization of shared flesh, to the future unfolding of the enfolded past in the ritual entanglement of an elemental re-collection" (Keller 2017, p. 130). Working with Calvinism,

---

2　I am heavily indebted to Enrique Dussel's account on the life of Bartolomé de las Casas. See (Dussel 2003, pp. 41–43).
3　For detailed discussions on Eucharistic theology (symbolism versus real presence), see (Bieler and Schottroff 2007, pp. 127–55).

Ernst van den Hemel echoes Keller's emphasis of the fold in his exposition on *extra calvinisticum*. Van den Hemel's article responds against the simplistic argument that highlighting the materiality of the Eucharist might lead to its veneration—idolatry of some sort. The divine's relationality with the material has been a contentious issue because it could be a slippery slope between venerating matter on the one hand, or secluding the presence of Christ purely in the abstract, on the other. According to van den Hemel, John Calvin's response to the heated debate on the Eucharist's transcendence and immanence during the sixteenth-century is also known and later coined as *extra calvinisticum*. The theory of the "extra," as van den Hemel posits, "leads the way to a view of the material world as inviting direct human interaction, rather than being governed purely by principles accessible to reason, like the laws of physics and systematic theology" (Van den Hemel 2012, p. 63). This dynamic response to the conundrum complements the perspective that presents the entanglements of the ungraspable divine with the solidity of the material. This article intensifies the "extra" by arguing not only for the descending and the union of the transcendent with the immanent, but also in the ontological reconfiguration of the transcendent as they are affected by the immanent.

This article's engagement with liberatory approaches to the Eucharist, however, does not negate the value of its liturgical expressions. Rather, this article acknowledges liturgy as its foundation and partner. As mentioned above, the Eucharist is a form of thanksgiving based on the Passover meal, a "sacrament of the gift which God makes with us in Christ through the power of the Holy Spirit" (World Council of Churches 2007, p. 10). This sacramentality is also a celebration of the "renewal of the covenant that God made with the earth and all living things" (Hill 1998, p. 143). As Brennan Hill suggests, an eco-justice approach to the Eucharist emphasizes its rootedness to the sacramentality of matter: "Bread made from wheat, and wine made from grapes . . . It views matter as a gifted part of creation and is strongly committed to this reality of the incarnation and the sacramental principle. Jesus was truly human and comes to us through the things of creation" (Hill 1998, p. 143). Margaret Scott even calls this sacramental mattering as "autobiographical:"

> The story of the bread and wine, like our own story and that of humanity itself, includes a painful chapter. It is a chapter that tells of oppression and death. The grain must be grounded into flour and the dough kneaded to become bread; the grapes must be crushed and trodden underfoot to become wine. In that bread and wine, the death and dying of so many people in our world every day, precisely through a lack of food and drink, is placed on the paten and poured into chalice. (Scott 2009, p. 51)

This article is my contribution to further reflect on the sacramentality of the eco-justice approach to the Eucharist. It participates by reimagining the ontology of the messiah who celebrated this feast. If the divine is commonly interpreted as transgressing to the material/creation in the Eucharist, then could the material/creation also have transgressed back (affectively, ontologically, and even materially) to the divine? In other words, could we imagine an ontological reconfiguration of the nature of the messiah by the bread and the wine?

## 2. Vitally Materializing the Elements

I limit this article's discussion of the Eucharist with the Gospel of Mark's version (Mark 14:22–25).[4] Although the Eucharist is attested four times in the New Testament (Matthew 26:26–29; Mark 14:22–25; Luke 22:15–20; 1 Corinthians 11:23–26),[5] only two (Luke and 1 Corinthians) out of the four instances bespeak of commemorating this event ("Do this in remembrance of me"—anamnesis[6]). Limiting the

---

[4]　This article primarily employs NRSV.

[5]　For comprehensive introductory texts on the Eucharist and the New Testament, see (LaVerdiere 1991; Kodell 1988; Moloney 1990; Ossom-Batsa 2000).

[6]　After thoroughly marshalling various instances of its usage, Richard J. Ginn defines anamnesis as "remembrance." See (Ginn 1989, p. 19). For a more political and praxis-inclined understanding of anamnesis, see (Morrill 1989).

discussion to Mark's iteration of this event, this article highlights Mark's omission of the command to remember as an important christological move where the focus shifts from the act of remembrance to the reconfiguration of the ontology of the Markan Jesus.[7] After reconfiguring his body as bread (14:22), Jesus did not continue his speech with any command to remember this ritual/event/moment. Rather, after reconfiguring his blood (of the covenant) with the wine or fruit of the vine (14:24), Jesus declares that he will "never again drink of the fruit of the vine until that day when I drink it new in the kingdom of God" (14:25). It seems that this last declaration implies that this event was to never happen again, or at least with Jesus, until the kingdom of God arrives/happens.

When it comes to methodology, first I invoke Stephen D. Moore's post-poststructuralist reading of the Johannine Jesus in reconfiguring the messianic ontology of the Markan messiah. Although Moore worked with a different gospel, he provides various "nonhuman" assemblages of the Johannine Jesus beyond those that are ordinarily adduced. Moore's description of the Johannine Jesus as "a vegetable byproduct" ("I am the bread" [John 6:35; 6:41,48,51]) (Moore 2017, p. 107) yields a post-poststructuralist reading that is applicable in engaging the ontology of the Markan messiah particularly in the Eucharist scene (Mark 14:22–25).

As Moore points to the human-nonhuman assemblages of the Johannine Jesus, the Markan Jesus, particularly in Mk 14:22–25, is also an assemblage of heterogenous matters. Tracing its definition to Deleuze and Guattari, Moore defines assemblage as "a configuration of heterogeneous elements that enter into temporary relations with one another and produce affects, effects, and entire realities" (Moore 2017, p. 114).[8] Assemblage is a philosophical concept that questions anthropocentric tendencies for human uniqueness and superiority. To combat this tendency, assemblage approaches ontology as fluid or in affective relationality with all entities. This fluidity includes, perhaps even focuses, on deconstructing identity as always influenced, changed, and composed by all entities. In other words, humans are never just humans. Our identity is a mixture of various elements that surrounds us, which includes animals, plants, and even inanimate objects. For example, food is something that is "nonhuman" and yet becomes more-than-human when they are ingested. In a certain sense, they determine the mood of the eater and even shape their form. Inasmuch as anthropocentrism tries to classify inanimate entities such as food as inferior compared to humans, their affective capacity in many cases are much more than those who have digested them.

Working with Jane Bennett's definition of vital materialism,[9] I consider the bread and the wine of the Eucharist as actants who vitally affect, influence, and change the other with their material presence. Coined by Bruno Latour, Bennett defines actant as

---

7　This shift assumes that the author of Mark knows the Eucharistic traditions and their liturgical expressions. This assumption may include Mark's knowledge of 1 Corinthians 11:23–26's incorporation of anamnesis in its recollection of the Last Supper. Other scholars argue that there is a possibility for a shift in framework in explaining Mark's lack of anamnesis. This shift in framework is explained as follows: "We have moved some way from the framework of the Jewish meal and its graces, perhaps to a purely gentile milieu. Those who hold that Mark has preserved the original tradition have to look elsewhere for their models: To 'the blessed bread of life,' 'the blessed cup of immortality, and 'the chrism of incorruption' in *Joseph and Aseneth*, to the Essene meals as described by Josephus (*Wars* II, pp. 139–43), or to the solemn formal meals at Qumran, of which none may partake until the priest has blessed the bread and the wine" (Jones and Hickling 1992, p. 196). Kodell concisely summarizes various Jewish meals that could be the model for the Eucharist (ex: daily Jewish meals, festive meals, Passover meal, Kiddush, Haburah, Essene meal, and Todah)—see (Kodell 1988, pp. 38–52). Joachim Jeremias even mentions the possibility of tracing this event to third-century Christian groups, the Quartodecimans (celebrates Eucharist by following the Jewish calendar in which it is celebrated on the 14th day of Nisan)—see (Jeremias 1966, pp. 207–18).

8　Manuel DeLanda defines assemblage or *agencement* "to the action of matching or fitting together a set of components (*agencer*), as well as to the result of such an action: An ensemble of parts that mesh together well." See (DeLanda 2016, p. 1). Other references/definitions on assemblage are found here: (Deleuze and Parnet 2002, p. 69; Deleuze and Guattari 1987, pp. 34, 38, 67, 73, 88, 90, 97–98, 323–24, 330, 356–57, 368, 503; Deleuze and Guattari 1994, p. 36; Guattari 2011, pp. 47, 55, 147, 188 and Guattari 1996, pp. 154–55).

9　"By 'vitality' I mean the capacity of things—edibles, commodities, storms, metals—not only to impede or block the will and designs of human but also to act as quasi agents or forces with trajectories, propensities, or tendencies of their own. My aspiration is to articulate a vibrant materiality that runs alongside and inside humans to see how analyses of political events might change if we gave the force of things more due" (Bennett 2010, p. viii).

A source of action that can be either human or nonhuman; it is that which has efficacy, can do things, has sufficient coherence to make a difference, produce effects, alter the course of events. It is "any entity that modifies another entity in a trial," something whose "competence is deduced from [its] performance" rather than posited in advance of the action.[10]

The bread and the wine of the Eucharist are not just food; they are (consecrated) actants. Human hands (or other devices) might have picked them up and directed them to (human) digestive organ; but, the Eucharistic elements are the ones who generate affectivity. To eat food/elements is/are "to enter into an assemblage in which the 'I' is not necessarily the most decisive operator" (Bennett 2010, p. 40). Food nourishes and produces energy/life. Yet, it also debilitates and even kills. Like *matryoshki* dolls, as Bennett describes, a human body is an assemblage that "contain a sequence of ever small ones—functioning groupings of actants in a series of larger, more complex congregations" (Bennett 2010, p. 45). These "ever small ones" include food. Their emergence in the human body assemblage has the capacity to change the larger assemblage: They are "assemblage converters": (Bennett 2010, p. 42) They have the power to alter direction and decisions, to irritate, and even to desensitize (food coma).

In most cases, the bread and the wine of the Eucharist in modern liturgical expressions are taken in fraction. Such limitation makes it hard to fully experience the actancy of the elements/food aside from those with celiac disease. Nevertheless, one has to re-imagine the meal that Jesus partook with his disciples, as attested in Mark 14:22–25, not as having a dinner spread the size of a coin and a sip/dip. Instead, it could have been a feast bountiful enough for thirteen adult men to eat. As Jesus ingests the bread and the wine, his body assembles with them. By doing so, Jesus' body reterritorializes into a new assemblage with the bread and wine.

Unlike Paul, Mark does not have robust exposition on the body (*sōma*). The Greek word *sōma* is attested only four times in Mark: Three times for the body of Jesus (14:8, 22; 15:43) and one time for the woman with excessive flow of blood (5:29). None of them provide critical exposition on the body/*sōma* as Paul has done in 1 Corinthians 6 and 15, particularly in contrast/relationship to flesh/*sarks*. Morna Hooker also limits Mark's understanding of *sōma* as the physical body, possibly "person or self."[11] In other words, the body/*sōma* that I discuss in this article engages the physical body of Jesus. Even if 1 Corinthians is written earlier than Mark, for whatever reason, I do not see Paul's influence in Mark's understanding of the body. Such distancing enables the possibility to understand the body of the Markan messiah as (processed) vegetal—bread and wine—without having to go through the complex Pauline classifications (or their fluidity) of the body-flesh discussion (Rivera 2015; Martin 1995; Engberg-Pedersen 2010; Asher 2000).

For Jesus to signify himself with the bread and the wine is to reconfigure himself into an "edible matter."[12] He is now an actant to be ingested by the other. But why did he choose to be identified with something that is ingested by the other? According to Monika K. Hellwig, "hunger brings into focus human dependence on the bounty of nature, and raises the important question about the source of that bounty and our relationship to it. But hunger also brings into focus the human person's dependence on other human beings" (Hellwig 1976, p. 15). For Jesus to become food, or at least related to food, is for him to teach his disciples that the messiah is part of this matrix of interdependence particularly of those who are hungry and suffering. As the feeding of the multitude (6:30–44 and 8:1–9)[13] reflects Mark's preference to depict his messiah as responding to the physical needs of the people (Bieler and Schottroff 2007, p. 77), Mark's depiction of his messiah in 14:22–25 is responding to their ontological hunger. Assuming that the Gospel of Mark was written shortly after the destruction of the second

---

[10]　From (Bennett 2010, p. viii). Bennett quotes (Latour 2004, p. 237).
[11]　(Hooker 1991, p. 341). Bieler and Schottroff echo Hooker's sentiments: "The Greek word *sōma* refers to the physical body, not to a bodily existence outside the present reality" (Bieler and Schottroff 2007, pp. 62–65).
[12]　This expression comes from Moore's description of the Johannine Jesus as an edible matter. See (Moore 2017, p. 117).
[13]　The feeding narratives allude to the manna received by the Hebrews in the wilderness. See Exodus 16:13–35.

temple of Jerusalem,[14] this hunger then reflects Mark and his community's desire to know the messiah and his kin(g)dom/Empire in the midst of this traumatic event. Perhaps, the so-called failure of the disciples to understand the Markan messiah is a form of hunger that is somewhat nourished by the (processed) onto-vegetal sustenance provided by Jesus at his last meal.

Through his edibility, we also become aware of the importance of all the vegetal lives that have been processed for our nourishment. Their salvific sacrifice is uplifted by no other than the messiah. Angel F. Méndez Montoya expresses her version of this ontological reconfiguration as "gastroeroticism:"

> Eucharist allows the partaker to enter into a deeper unity between the human and the divine, the immanent and the transcendent. Matter is divinized, but only through its own materiality. Humanity is deified but only in the midst of its situatedness. The bread and the wine become Christ's body and blood, but without setting aside their edible characteristics of bread and wine. In this sense, transubstantiation is not a mere extrinsic act but intrinsic, a radical expression of divine intimacy and love enacted by the Holy Spirit from within creation and at the core of human flesh. The gastroeroticism that takes place in the Eucharist is this divine desire-love already nourished from within the situatedness of the particularities of creation and local communities. (Méndez Montoya 2009, p. 72)

Méndez Montoya expresses the Eucharist in its "*metaxu* or in-betweenness: God and creation, transcendence and immanence, word and action, desire and satiation, eros and agape, self and other" (Méndez Montoya 2009, p. 159). Although she does not explicitly engage the in-betweenness of (the Markan) Jesus' ontology, one can find traces from her work the possibility for a messiah whose in-betweenness is also ontological. That is, the Markan Jesus is an assemblage of human and nonhuman metaxologically. Here, Méndez Montoya does not neglect to say that this in-betweenness includes and even highlights humans' interdependence with all creations.

Yet, this turn to edibility "risks of dehumanization" (Moore 2017, p. 118) as Moore qualifies. The Markan Jesus's body, a colonized body, was turned into a lifeless corpse at the cross, consumed by the gaze of its oppressors (15:22–41). In all its gruesome "thingification"[15] or the treatment of certain bodies as dispensable, the reconfiguration of the Markan messiah as an assemblage of human and nonhuman cannot simply ignore the history of animalization against oppressed and minoritized bodies.

Moreover, this article could be accused of ventriloquizing plants with my own human voice. Michael Marder's critique against the Eucharist echoes the unhindered appropriation of the body of plants as sacrificial ingredient(s) for salvific consumption. Processing plants for human consumption (literally and symbolically), according to Marder, destroys their ontological possibilities:

> Through the sanctified human activities of cultivating certain kinds of plants and transforming them into edible or drinkable substances (here, I repeat, we are dealing with a very telling example), the subaltern plant, itself incapable of speech, is represented by and commences to speak with more than one voice and in more than one tongue: it comes to ventriloquize at once the voice of Reason and that of Revelation . . . and so ceases to be a plant. (Marder 2013, p. 33)

This instrumentalization of plant life is not unheard of in Mark. In the infamous cursing of the fig tree narrative (11:12–14, 20–21), Mark depicts his Jesus as "someone who was influenced by the colonial and anthropocentric desire to interfere, manipulate, and commodify plants" (Jeong 2018, p. 114). Cursing the fig tree reflects Jesus' mimicry of the "oppressive temporality that demanded un-vegetal ripeness or the commodification of plant time" (Jeong 2018, p. 114). In the same way, the Eucharist could be misconstrued as instrumentalizing plants by processing them materially and

---

14  LaVerdiere alternatively frames it in the context of Nero's persecution of the followers of Jesus at Rome. See (LaVerdiere 1991, p. 48).

15  Moore quotes Aimé Césaire's use of this expression (Moore 2017, p. 118). See (Césaire 2000, p. 36).

symbolically for human (sacramental) consumption. Even before encountering, attending, and being responsive to the plants, their ontology could have already been exhausted for anthropocentric ends.

Echoing Marder's concern, the Australian Catholic Social Justice Council questions the uncomfortable inquiry about the cost of growing the bread and the wine used in the Eucharist (Scott 2009, p. 61). Do the bread and the wine served in our churches support or condone oppressive, capitalistic structures that abuse human labor and the earth? What is the cost of consecrating the body and blood of Jesus? Have we inadvertently sacrificed the poor of the world (humans and nonhumans) by replacing their centrality in understanding the Eucharist with elaborate garments, patens, and chalices?

If Marder demands a vegetal resolution that does not drive plants to their ontological exhaustion, then the Eucharist should also not be understood in limited terms. Instead, the Eucharist should also be accounted for its infinite possibilities, expressions, and essential incompletions. Ecologically concerned celebration of the Eucharist does not oppose, as Marder's plant-thinking suggests,

> The use of fruit, roots, and leaves for human nourishment; rather, what it objects to is the total and indiscriminate approach to plants as materials for human consumption within the deplorable framework of the commodified production of vegetal life. If one is to respect vegetal existence, one will facilitate, not restrict, the proliferation of its various ends, not to mention celebrate the lack thereof. (Marder 2013, p. 184)

In other words, Marder's ontophytology or understanding ontology from/with plants provides a way to perform the Eucharist that is responsive to the plants. That is, Marder invites us to ask: "How am I to eat ethically?"

> To put it succinctly, if you wish to eat ethically, *eat like a plant*! Eating like a plant does not entail consuming only inorganic minerals but welcoming the other, forming a rhizome with it, and turning oneself into the passage for the other without violating or dominating it, without endeavoring to swallow up its very otherness in one's corporeal and psychic interiority. (Marder 2013, p. 185)

To ontologically reconfigure the Markan messiah as an assemblage of humans and nonhumans is "to eat like a plant" or to be in rhizomatic[16] connection with all of creation. Such connection demands caring for the other not just in the plane of charity (or handouts) but constant (re/de)-territorialization[17] of the ontological with the other. This territorialization is first to admit that we cannot completely live harmlessly or without depending upon the other for food. Second, as Wendell Berry suggests, "we do this knowingly, lovingly, skillfully, reverently, it is a sacrament" (Berry 1981, p. 281). The sacramentality of the Eucharist, aside from all of its theological expositions, is found when we, like Jesus, reconfigure our ontology with the (processed) plants. To consecrate is to become (processed) vegetal.

## 3. Vegetal (Rice) Covenant

If Matthew's version of the Eucharist explains that the blood of the covenant is poured out for many so that their "sins could be forgiven" (26:28), Mark does not have this explanatory clause. Such lack does not completely steer Mark's Eucharist away from the ancient Jewish tradition and

---

16　"Six principle characteristics of the rhizome: (1) Connection (vs. order or model), (2) heterogeneity of coding, where semiotic chains connect other assemblages; (3) multiplicity in determination, magnitude, or dimension (vs. unity in subject or object), (4) a-signifying ruptures of segmentation, stratification, and territory, (5) cartographic production (vs. tracing), and (6) 'decalcomania,' in that any tracing (as with a decal that is transferred onto another medium) would be in fact 'be put back on the map' because apparent reproduction gives way to asymmetry or difference" (Young et al. 2013, p. 262).

17　DeLanda defines territorialization as referring "not only to the determination of the spatial boundaries of a whole—as in the territory of a community, city, or nation-state—but also to the degree to which an assemblage's component parts are drawn from a homogenous repertoire, or the degree to which an assemblage homogenizes its own components" (DeLanda 2016, p. 22; Deleuze and Guattari 1987, p. 315; Young et al. 2013, pp. 306–12).

remembrance when the Israelites were sprinkled with the blood of the covenant which YHWH has made for them in Sinai (Ex 24:4–8; Zech 9:11–15 recalls this event).[18] However, such lack creates room for, as Bieler and Schottroff suggest, "God's mercy, the mercy God promised to the ancestors, the expression of the promise that the people will have a great progeny and a future."[19] The Markan covenant is not a replacement of the Sinaitic version. Rather, it is a continuation of God's mercy and promise to bring hope in the midst of their suffering. From a vital materialist and vegetal perspectives, this hope is manifested non-anthropocentrically. This covenant, which is poured out for many, resides and is found in the (processed) vegetal. In particular, it is found in the (processed) vegetal body and blood of the Markan Jesus.

I read Mark's Eucharist through the lens of vital materialism because as Tat-siong Benny Liew suggests, "What if biblical materials, in separation from the question of faith or belief, turn out to have instructive bearing on some of the urgent issues facing our world today?" (Liew 2008, p. 54). For this article, I find the unchecked destruction of the earth by human hands as one of the urgent issues of our time. So, I approach Mark 14:22–25 as a timeless response to our world, hoping to be stirred by the vitality of understanding the Markan messiah as a (processed) vegetal who is in solidarity, continuing God's mercy, with all of creation.

This "timelessness" and in connection with the wider "world" also bespeaks of the urgent need to contextualize the Eucharist in a globalizing world. If this vital materialist reading of Mk 14:22–25 finds the possibility for the bread and the wine to ontologically affect the Markan messiah, then an Asian, Korean in particular, contextual approach to this affective reconfiguration prefigures the possibility for the Markan messiah to be affected by other (processed) vegetal entities as well, namely rice. Like the intense (often times oppressive) labor that goes into the production of bread and wine in the Mediterranean region, Soonwhan Kim highlights the sacrificial, bitterly painful labor of the (Korean) rice farming culture (Kim 2002, p. 152). Throughout the year, farmers "pour out their sweat, blood, and life into planting, raising, reaping, and threshing rice" (Kim 2002, p. 152). They do so because rice is life: "It always functions both as a sociological and as a religious medium that reflects the devotional heart and mind of the people" (Kim 2002, p. 151). This reminds Kim of the sacrificial and life-giving gift Jesus has provided for all creation. Moreover, rice/nature is not subordinate to humans; rather, rice/nature are in "symbiotic partnership" with humans (Kim 2002, p. 154). Kim argues that a proper relationship with nature is for "humans to adjust to the inclinations of nature, not the other way around" (Kim 2002, p. 154). Although Kim does not utilize nonhuman or eco-justice concepts explicitly, his work opens the possibilities for a contextual (Korean) vital materialist reading of Mk 14:22–25 in which the Markan messiah ontologically reconfigures his identity as the messiah with rice. This reconfiguration with rice should not be engaged with historicist inquiries in which the existence/use of rice during the time and place of Mark/Jesus is in question. Rather, it acknowledges that the very act of reconfiguring the messianic body/ontology with the other is Mark's way of inviting his readers/listeners that his messiah's identity is with those who partake his body and blood, whenever, wherever, and whoever they are. This fluid relationality on the one hand does not limit the kinds of ordinary [bread, wine, rice, *makgeolli* (Korean rice liquor), *puto* (rice cake), *lambanog* (coconut wine), and others]; on the other, it insists to remember the gift of life given by the (poor) farmers and the vegetal bodies.

**Funding:** This research received no external funding.

**Conflicts of Interest:** The authors declare no conflicts of interest.

---

[18]  (Van Iersel 1998, p. 426). The shedding of blood is found in multiple passages that use it for slaughtering sacrificial animal. For example: Ex 29:12; Lev 4:7,18,25,30,34; 8:15; 9:9; 2 kings 16:15. cf. (Van Iersel 1998, p. 426).
[19]  See Jeremiah 31:31–34. cf. (Bieler and Schottroff 2007, p. 61).

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
