# Peer review of "The (Processed) Vegetal Body and Blood of the Markan Messiah"

_religions, doi:10.3390/rel10010001_

Reviewer 1 Report

This is an interesting, engaging and original contribution to the Religions special issue on 'Verdant: Knowing Plants, Planted Relations, Religion in Place'. The author argues that 'the Markan Jesus is an assemblage of the human and the (processed) vegetal body' (p. 2). Critical plant studies and new materialism are combined to offer a new understanding of the Eucharist as the agentic, intercorporeal assemblage of Christ and vegetal bodies.  

Some minor revisions and the consideration of a few conceptual points are as follows.

The article foregrounds the processed vegetal body, particularly in the section 'Vitally Materialising the Elements', but the critical difference between 'processed' and 'devitalised' plant (wheat and grape) bodies remains unclear. This point is especially important to clarify in the context of GMO plant varieties and the devitalisation/denaturation of mass produced agricultural plants. How might the consecrated body of Christ co-vitalise with the potentially denatured bodies of mass agriculture?

Secondly, I believe an eco-justice of plants would also need to recognise their deaths--their giving over of their lives to the sacramental body. On vegetal death and mourning, for example, see John Ryan (2018) Plants in contemporary poetry, chapter 8 'On the death of plants'. To negate the sacrifice made by plants -- and which we make of them -- could be to risk anthropocentrism and vegetal blindness.

Thirdly, I'm very curious to hear more about food coma (desensitisation) and celiac (gluten intolerance), as these sorts of conditions would impede the transubstantiation of human-plant-sacred bodies.  

Edits:

The MDPI referencing system would need to be applied, should the article be published. Currently, the article makes use of in-text parenthetical references.

p. 2: ‘…then the bread and the win of the Eucharist become… (change ‘becomes’ to ‘become’)

p. 2 ‘The bishops are to care for the “the orphans…’ (remove first ‘the’)

p. 4: ‘…transgressing to…’ (remove ‘to’ here and elsewhere)

p. 6: ‘food’ is treated as a plural noun throughout these paragraphs. Please reconjugate sentences so that food is an ‘it’ rather than a ‘they’. Otherwise, this section reads awkwardly. 

Author Response

Thank you again for the opportunity to submit my work in your journal.

My article is titled: “The (Processed) Vegetal Body and Blood of the Markan Messiah.”

Please see below the edits I made that fulfill the reviewers’ suggestions:

1) The title has been changed from “in Mark’s Eucharist” to “of the Markan Messiah”

2) Neruda’s epigraph is deleted.

3) Lines 27-41 are added/edited.

Thank you very much for the helpful suggestions.